# Impact of intensive adherence counseling on viral load suppression and mortality among people living with HIV in Kampala, Uganda: A regression discontinuity design

Jonathan Izudi [1,2,3]*, Barbara Castelnuovo[1], Rachel King[4], Adithya Cattamanchi[5,6]

1 Infectious Diseases Institute, College of Health Sciences, Makerere University, Kampala, Uganda,
2 University of California Global Health Institute (UCGHI), University of California, San Francisco, San Francisco, CA, United States of America, 3 African Population and Health Research Center (APHRC), Nairobi, Kenya, 4 Institute for Global Health Sciences, Department of Epidemiology and Biostatistics, University of California, San Francisco, California, United States of America, 5 Center for Tuberculosis, San Francisco General Hospital, University of California, San Francisco, San Francisco, CA, United States of America, 6 Division of Pulmonary Diseases and Critical Care Medicine, University of California, Irvine, Irvine, CA, United States of America

* jonahzd@gmail.com

**Data Availability Statement:** All relevant data are within the manuscript and its Supporting Information files.

## Abstract

Intensive adherence counseling (IAC) is recommended for people living with HIV (PLHIV) with viral load (VL) $\geq$1,000 copies/ml after $\geq$6 months of anti-retroviral therapy (ART). We evaluated the effect of IAC on VL suppression and all-cause mortality among PLHIV on first-line ART with VL $\geq$1,000 copies/ml after $\geq$6 months of ART in Kampala, Uganda using regression discontinuity design, a quasi-experimental method for effect estimation when interventions depend on a cut-off. PLHIV just above VL $\geq$1,000 copies/ml cut-off who received $\geq$3 IAC sessions formed the intervention group while those just below the cut-off who received routine psychosocial support constituted the control group. Primary outcome was repeat VL suppression defined as VL <1,000 copies/ml approximately 9–12 months following initial VL assessment. Secondary outcome was all-cause mortality. We used logistic regression for causal-effect analysis, reported as odds ratio (OR) with a 95% confidence interval (CI). We performed sensitivity analyses to assess the robustness of findings to varying bandwidths at the cut-off. We found 3,735 PLHIV were started on ART between Nov 2020 and Nov 2021 of whom 3,199 were included in the analysis (3,085 control, 114 intervention). Within an optimal bandwidth, there were 236 participants (222 control, 14 intervention) with similar demographic and clinical characteristics. Repeat VL suppression was lower in the intervention than in the control group (85.7% versus 98.6%, p = 0.021) while all-cause mortality was similar (0% versus 0.5%, p = 1.000). In multivariable analysis, the odds of repeat VL suppression were 91% lower in the intervention than control group (OR = 0.09; 95% CI, 0.01–0.66). Findings are robust to varying bandwidths around the cut-off. We concluded IAC is ineffective in suppressing VL among PLHIV on first-line ART in Kampala, Uganda. Findings suggest a need to investigate the IAC implementation fidelity for successful translation in practice and the reasons for VL persistence beyond the suppression threshold.

**Funding:** This project was supported by the Fogarty International Center of the National Institutes of Health (NIH) under Award Number D43TW009343 and the University of California Global Health Institute (UCGHI) to JI. The content is solely the responsibility of the authors and does not necessarily represent the official views of the NIH or UCGHI. The funders had no role in study design, data collection and analysis, decision to publish, or preparation of the manuscript. The Kampala Capital City Authority (KCCA) HIV clinics are supported by funding from the Government of Uganda (GR-G-0902) and the President's Emergency Plan for AIDS Relief through the United States Centers for Disease Control and Prevention (co-operative agreement NU2GGH002022).

**Competing interests:** The authors have declared that no competing interests exist.

## Introduction

With optimal adherence to Anti-Retroviral Therapy (ART), nearly all people living with HIV (PLHIV) achieve viral load (VL) suppression (VL <1,000 copies/ml) within six months of ART initiation [1]. VL suppression reduces vulnerability to opportunistic infections, improves the quality and quantity of life of PLHIV, reduces the risk of HIV transmission during sexual intercourse, prevents mother to child transmission of HIV during labor, delivery, and breast-feeding, and it is indicative of an optimally performing HIV control program [2]. The World Health Organization (WHO) recommends intensive adherence counseling (IAC) for those with unsuppressed VL (VL ≥1,000 copies/ml). IAC consists of three sessions of targeted and structured counseling and support, provided one month apart by a multidisciplinary team to help PLHIV with unsuppressed VL overcome barriers to ART adherence [1].

Existing evidence regarding the effectiveness of IAC is mixed. A systematic review and meta-analysis reported that 46.1% of PLHIV had VL suppression after three IAC sessions [3], with VL suppression being lower among children and adolescents than adults [3]. A retrospective cohort study conducted in Zimbabwe reported 1–2 or ≥3 IAC sessions are associated with a higher likelihood of VL suppression than a single IAC session [4]. In Uganda, a prospective cohort study involving PLHIV on long-term ART with unsuppressed VL found less than one in 10 had VL suppression after completing IAC [5], suggesting IAC is ineffective in reversing unsuppressed VL. In children and adolescents living with HIV in Uganda, less than a quarter suppressed their VL after three IAC sessions [6].

While studies conducted on IAC and VL suppression in Uganda and elsewhere appear to suggest that IAC has little to no effect on VL suppression, the causal effect of IAC on VL suppression is unknown. One study used a before and after design [7] while several others used a retrospective cohort design [4, 6, 8, 9] to measure the effect of IAC on VL suppression, therefore, precluding an assessment of the causal effect of IAC on VL suppression. The EFFectiveness of INtensive Adherence Counselling (EFFINAC) study was designed to evaluate the causal effect of IAC on VL suppression and all-cause mortality among adolescents and adults on first-line ART with unsuppressed VL after six or more months of ART in Kampala, Uganda using real-world (programmatic) data.

## Methods and materials

### Study setting and data source

The EFFINAC study was conducted at six government health facilities in the Kampala metropolitan area in Uganda. All the study sites were primary healthcare units and they included Kisenyi, Kawaala, Kisugu, Kitebi, Komamboga, and Kiswa Health Centers. Kisenyi is a Health Center (HC) IV or a county-level health facility serving nearly 100,000 people and the others are HC III or parish-level health facilities and each serves approximately 20,000 people [10]. Each of the study sites has an HIV clinic that provides HIV prevention, care, treatment, and support to PLHIV according to the Uganda National HIV/AIDS Treatment Guidelines [1]. The clinics collect HIV data at each patient visit using the Uganda Ministry of Health (MoH) standardized HIV/ART card. Collected data are entered into an electronic medical record system. We retrieved, cleaned, and analyzed the data (S1 Data). The health facilities receive technical support and assistance in data management from the Infectious Diseases Institute (IDI). Further details about the study sites are described elsewhere [11, 12].

## Ethical issues

The EFFINAC study received ethical approval from the IDI Research Ethics Committee (approval #IDI-REC-2022-18) and the Uganda National Council for Science and Technology (approval #HS25553ES). Administrative clearance was obtained from the Directorate of Public Health and Environment of KCCA (approval # DPHE/KCCA/1301). Since existing medical records were retrieved and analyzed, the requirement for informed consent was waived by the IDI-REC and UNCST.

## Inclusivity in global research

Additional information regarding the ethical, cultural, and scientific considerations specific to inclusivity in global research is included in the S1 File.

## Study design

We used a regression discontinuity design (RDD), a quasi-experimental approach to measure the causal effect of IAC on VL suppression and all-cause mortality. RDD is appropriate when intervention assignment depends on a cut-off [13–15].

The provision of IAC depends on VL being ≥1,000 copies/ml so that PLHIV above the cut-off receive IAC while those below the cut-off continue with routine psychosocial support (PSS). An RDD study simulates a randomized controlled trial at the cut-off (local randomization) [16] by comparing outcomes among participants who are just above and just below the cut-off since the two groups are likely to be similar in both measured and unmeasured confounders [17]. With IAC, the probability of assignment to IAC shifts from zero to one at the cut-off, a phenomenon termed discontinuity. The conditions for an RDD include an assignment or forcing variable measured on a continuous scale, a cut-off, and outcome(s) measured for all participants regardless of the intervention [14]. In the EFFINAC study, the assignment variable was VL following six or more months of ART, the cut-off was VL ≥1,000 copies/ml, and the outcomes were VL suppression and all-cause mortality.

## Study population

Between Dec 1, 2022, and Jan 5, 2023, we extracted data through Nov 1, 2022 from the electronic medical record system on all PLHIV who initiated ART at the six study sites between Nov 1, 2020 and Nov 30, 2021. Additional data for PLHIV receiving IAC were abstracted from the IAC register.

We included PLHIV aged ≥15 years who were initiated on a first-line ART regimen at one of the six study sites between Nov 1, 2020 and Nov 30, 2021. The HIV resistance level of the participants was not known as the testing is not universally performed due to its high cost. The participants were primarily on Dolutegravir (DTG)-based regimens and followed for ≥12 months until Nov 1, 2022 to allow full data collection on IAC and repeat VL. The first-line ART regimen in Uganda consists of the possible combinations of Tenofovir (TDF) or Abacavir (ABC), Lamivudine (3TC), and Dolutegravir (DTG), Efavirenze (EFV) or Atazanavir/Ritonavir (ATV/r). We excluded PLHIV on Non-Nucleoside Reverse Transcriptase Inhibitors (NNRTI) due to a higher level of pre-treatment resistance at 15.9% in this population per the Uganda National Antiretroviral Drug Resistance Committee study [1]. In the EFFINAC study, the intervention group included PLHIV just above the VL ≥1,000 copies/ml cut-off who received ≥3 IAC sessions. The control group consisted of PLHIV just below the VL ≥1,000 copies/ml cut-off who received routine PSS. In both groups, PLHIV with no repeat VL data and those transferred out to other health facilities were excluded.

## Measurements

**Intervention versus control groups.** IAC was the intervention and was used at all six study sites in accordance with the Uganda MoH HIV treatment guidelines. IAC refers to a psychosocial counselling and support intervention provided by a multidisciplinary team of health workers (clinicians, nurses, counsellors, family members, and peers among others) to PLHIV with unsuppressed VL after six or more months of ART. IAC is recommended when unsuppressed VL results from suboptimal ART adherence but not HIV drug resistance. In Uganda, HIV drug resistance testing is not routinely performed for PLHIV on a first-line ART regimen [6]. IAC aims to overcome the barriers to optimal ART adherence and usually three or more IAC sessions spaced one month apart and scheduled according to routine appointment dates are provided to PLHIV with unsuppressed VL.

IAC is provided according to five A's: 1) Assess: assessing adherence levels, progress in dealing with barriers, compliance to adherence plans, and new barriers to ART adherence; 2) Advice: advising on specific barriers to ART adherence, benefits of good ART adherence and the consequences of ART non-adherence; 3) Assist: assisting PLHIV in identifying barriers to ART adherence, prioritizing the barriers, developing possible solutions, and discussing the pros and cons of each solution; 4) Agree: agreeing on solutions to address key barriers, evaluating and documenting each solution, and developing a new ART adherence plan; and 5) Arrange: Here, the team summarizes the prioritized solutions, review the new ART adherence plan, explain the IAC schedule, emphasize appointment keeping, schedule for subsequent IAC sessions, arrange/or refill ART, and provide a return date. Expert consultations and referrals are sought to address complex issues, for example, stigma, disclosure, mental health challenges, and nutrition. All IAC sessions received are reported in the MoH IAC register to ease the tracking of the sessions per individual. After completing IAC ($\geq$3 IAC sessions), repeat VL testing is performed and PLHIV with VL $\geq$1,000 copies/ml are categorized as treatment failure and are switched to a second-line ART regimen according to the HIV treatment guidelines. However, PLHIV with VL<1,000 copies/ml after completing IAC are reverted to routine PSS to maintain their suppressed VL status. In the control group, health workers provide routine PSS to PLHIV with suppressed VL following six or more months of ART to help maintain VL suppression. At all six study sites, PSS is provided by individual health workers during routine HIV care at both the health facility and community levels. These health workers interact with PLHIV throughout the process of HIV care and treatment, including during health education, consultation, and ART refills.

Routine PSS services are provided according to the needs of PLHIV and may include any of the following: 1) psychosocial screening and assessment for adolescents using the Home, Education/ Eating/ Employment, Activity, Drugs, Sex, and Sexuality, Suicidal ideation/mental health (HEADSS) tool at each clinical visit; 2) general and targeted health education; 3) ART adherence preparation, monitoring, and support; 4) counselling and psychotherapy; 5) mental health screening using the Patient Health Questionnaire (PHQ)-2 tool followed by PHQ-9 tool if the score on the former exceeded three; 6) positive health dignity and prevention; 7) family and social support; 8) care and support for survivors of gender-based violence (GBV); 9) nutritional care and support; and 10) referral and linkage to specialized health care, socio-economic and legal support, including the fight against discrimination, and spiritual support. In the event of unsuppressed VL after PSS, IAC is provided as recommended.

## Baseline covariates

The health facility-related covariates included health facility characteristics, namely the name of the study site and level of care (HC III versus IV). Individual-level sociodemographic

covariates included age measured in absolute years and categorized as 15–24, 25–34, 35–44, 45–54, and 55 and more, and sex measured as female or male. Individual-level clinical covariates included the point of entry into HIV care, baseline WHO clinical stage (I, II, III, and IV), weight in kilograms, nutritional status measured based on mid-upper arm circumference (MUAC) categorized as well-nourished, moderate acute malnutrition and severe acute malnutrition, CD4 count (less or $\geq 500$ cells/ul), tuberculosis or TB status (no signs and symptoms of TB, currently on TB treatment, diagnosed with TB, TB treatment completed and presumed with TB). We classified ART regimens as DTG-based (TDF/3TC/DTG, TDF-FTC-NVP, TDF-FTC-EFV, TDF-3TC-NVP, and TDF-3TC-EFV), ABC-based (ABC/3TC/DTG and ABC-3TC-LPV/r), AZT-based ART regimen (AZT/3TC/DTG and AZT-3TC-NVP), and other first-line ART regimens.

## Study outcomes

The primary outcome was repeat VL suppression defined as VL <1,000 copies/ml measured after completion of IAC in the intervention group and after PSS in the control group. The secondary outcome was all-cause mortality, measured as death from any cause during the study period. Both outcomes were measured on a binary scale (yes or no).

## Statistical issues

The analysis was performed in R version 4.2.1 using the *"rdd"* and *"rdrobust"* packages and Stata version 15.0. We hypothesized there will be no difference in repeat VL suppression between the intervention and control groups. Two variables, MUAC and baseline weight, had missing observations which were imputed using the multivariate imputations via the chained equations (MICE). We conducted exploratory data analysis and graphical exploration of the validity of RDD.

Here, we summarized numerical data using the mean and standard deviation when normally distributed, or the median and interquartile range (IQR) when skewed. We computed frequencies and percentages for categorical data. We graphed the probability of treatment assignment, the outcomes, and the baseline covariates against the forcing variable to assess the type of RDD, the magnitude of effect, and covariate distribution, respectively at the cut-off. The distribution of observations at the cut-off was assessed by plotting the density of the forcing variable and testing using the McCrary test [18].

We tested differences in the distribution of baseline categorical variables such as sex across the intervention and control groups using the Chi-square test when the cell count was large ($\geq 5$) otherwise the Fisher's exact test was used for smaller cell counts (<5). The Student's t-test was used to test mean differences in numerical data such as age across the groups when the data were normally distributed else the Wilcoxon-rank sum test was used when the data were skewed.

We declared the dataset as an RDD object with a constant seed number to ensure reproducibility. We determined the optimal bandwidth (the lower and upper bound) at the VL $\geq 1,000$ copies/ml cut-off using the Imbens-Kalyanaraman method [14, 19] and extracted and saved the dataset for further analysis. To estimate the effect of IAC on the primary and secondary outcomes, we fitted a multivariable generalized linear model with a logit-link and binomial distribution with the outcomes as a function of IAC with and without adjusted for the baseline covariates and with and without restriction of the observations to the optimal bandwidth. We reported the results as odds ratio (OR) and the 95% confidence interval (CI).

### Sensitivity analysis and reporting of findings

We checked the robustness of the estimates using varying bandwidths, either increasing or decreasing at the VL ≥1,000 copies/ml cut-off. We adhered to the elements of Improving the Reporting Quality of Nonrandomized Evaluations of Behavioral and Public Health Interventions (The TREND) statement [20] in reporting the study results.

## Results

### Summary of study profile

Overall, we retrieved records on 3,735 PLHIV and excluded 536 (14.4%) for reasons: 512 participants had transferred to other health facilities while 24 were below 15 years of age (Fig 1).

### General characteristics of the participants

Among the 3,199 participants, mean age was 31.2 years (standard deviation, 9.5), 1,031 (32.3%) were from Kisenyi HC IV, 2,216 (69.3%) were female, and 1,941 (60.7%) were enrolled into HIV care through the Outpatient Department (Table 1). At ART initiation, 3,066 (95.8%) were in WHO clinical stage I, 3,025 (94.6%) were started on TDF/3TC/DTG (overall, 3,089 or 96.6% on TDF-based ART regimen) and 3,165 (98.9%) were well-nourished. The majority of

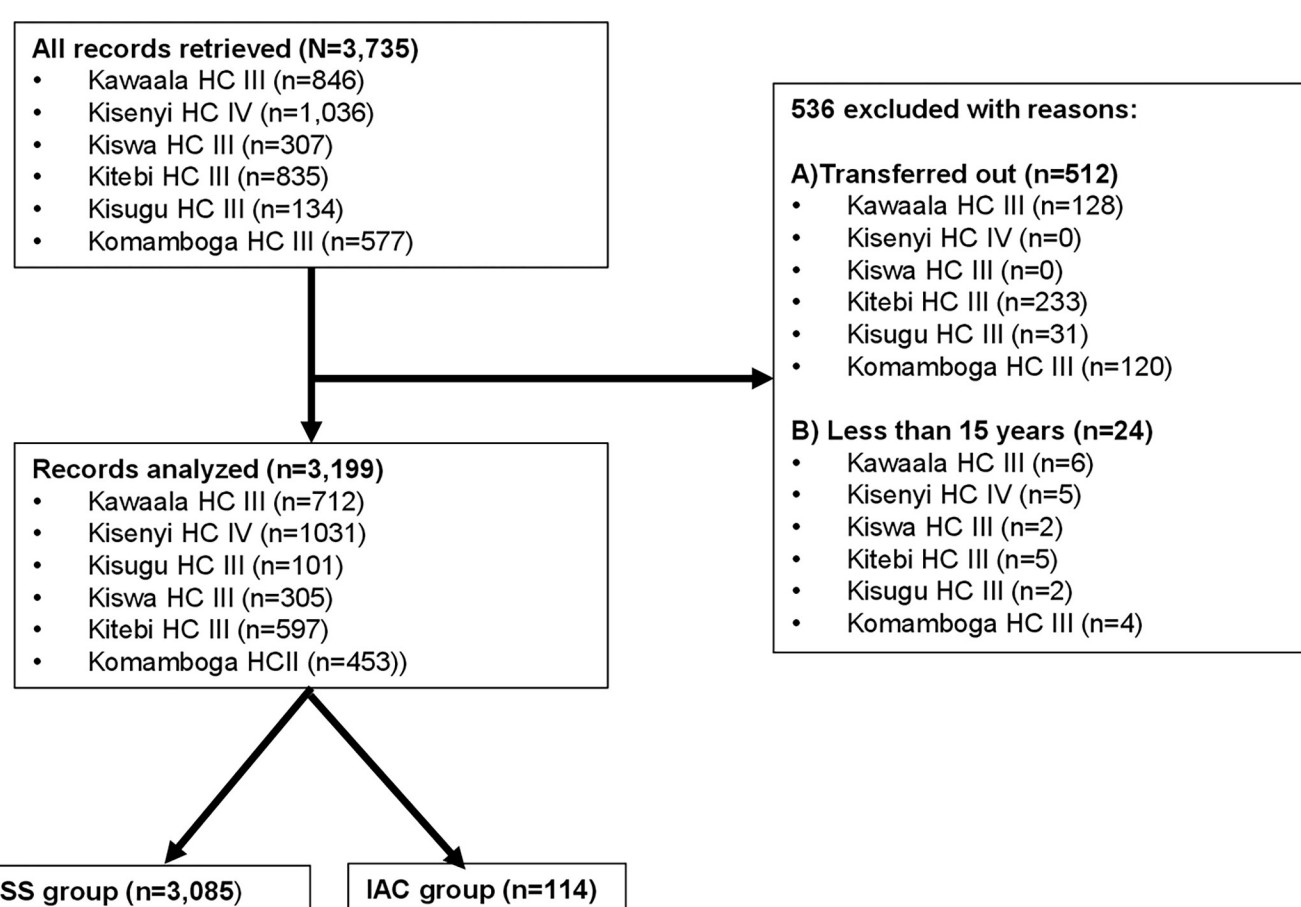

**Fig 1. Summary of EFFINAC study profile according to eligibility criteria.** Accordingly, we analyzed records for 3,199 participants: 3,085 (96.4%) in the control group and 114 (3.6%) in the intervention group. All 3,085 participants in the control group received routine PSS and all 114 participants in the intervention group received ≥3 IAC sessions.

**Table 1. Distribution of participant characteristics between the intervention and control groups before applying the optimal bandwidth.**

| Variables | Levels | Overall (n = 3,199) | Control group (n = 3,085) | Intervention group (n = 114) | P-value |
|---|---|---|---|---|---|
| Study site | Kawaala HC III | 712 (22.3) | 694 (22.5) | 18 (15.8) | 0.610 |
| | Kisenyi HC IV | 1031 (32.2) | 991 (32.1) | 40 (35.1) | |
| | Kisugu HC III | 101 (3.2) | 98 (3.2) | 3 (2.6) | |
| | Kiswa HC III | 305 (9.5) | 292 (9.5) | 13 (11.4) | |
| | Kitebi HC III | 597 (18.7) | 576 (18.7) | 21 (18.4) | |
| | Komamboga HCII | 453 (14.2) | 434 (14.1) | 19 (16.7) | |
| Level of health facility | HC III | 2168 (67.8) | 2094 (67.9) | 74 (64.9) | 0.573 |
| | HC IV | 1031 (32.2) | 991 (32.1) | 40 (35.1) | |
| Age categories (years) | 15–24 | 698 (21.8) | 658 (21.3) | 40 (35.1) | 0.001 |
| | 25–34 | 1481 (46.3) | 1428 (46.3) | 53 (46.5) | |
| | 35–44 | 751 (23.5) | 736 (23.9) | 15 (13.2) | |
| | 45–54 | 208 (6.5) | 205 (6.6) | 3 (2.6) | |
| | 55 and more | 61 (1.9) | 58 (1.9) | 3 (2.6) | |
| | mean (SD) | 31.2 (9.5) | 31.4 (9.4) | 27.1 (12.5) | <0.001 |
| Sex | Female | 2216 (69.3) | 2134 (69.2) | 82 (71.9) | 0.601 |
| | Male | 983 (30.7) | 951 (30.8) | 32 (28.1) | |
| Point of entry into HIV care | Outpatient Department | 1941 (60.7) | 1881 (61.0) | 60 (52.6) | 0.247 |
| | PMTCT Clinic | 799 (25.0) | 757 (24.5) | 42 (36.8) | |
| | Others | 232 (7.3) | 226 (7.3) | 6 (5.3) | |
| | TB Clinic | 168 (5.3) | 164 (5.3) | 4 (3.5) | |
| | Outreach clinic | 25 (0.8) | 25 (0.8) | 0 (0.0) | |
| | Young Child Clinic | 9 (0.3) | 8 (0.3) | 1 (0.9) | |
| | In-patient Department | 9 (0.3) | 8 (0.3) | 1 (0.9) | |
| | SMC Clinic | 7 (0.2) | 7 (0.2) | 0 (0.0) | |
| | Key Population Clinic | 5 (0.2) | 5 (0.2) | 0 (0.0) | |
| | Transfer-in | 2 (0.1) | 2 (0.1) | 0 (0.0) | |
| | EID Clinic | 1 (0.0) | 1 (0.0) | 0 (0.0) | |
| | STI Clinic | 1 (0.0) | 1 (0.0) | 0 (0.0) | |
| Baseline WHO clinical stage | I | 3066 (95.8) | 2962 (96.0) | 104 (91.2) | 0.016 |
| | II | 66 (2.1) | 60 (1.9) | 6 (5.3) | |
| | III | 52 (1.6) | 50 (1.6) | 2 (1.8) | |
| | IV | 15 (0.5) | 13 (0.4) | 2 (1.8) | |
| Baseline weight (Kgs) | mean (SD) | 60.89 (14.56) | 61.11 (14.41) | 54.95 (17.30) | <0.001 |
| Baseline ART regimen | TDF-based | 3089 (96.6) | 2994 (97.1) | 95 (83.3) | <0.001 |
| | ABC-based | 67 (2.1) | 53 (1.7) | 14 (12.3) | |
| | Other first-line | 38 (1.2) | 33 (1.1) | 5 (4.4) | |
| | AZT-based | 5 (0.2) | 5 (0.2) | 0 (0.0) | |
| Baseline MUAC | Well-nourished | 3165 (98.9) | 3052 (98.9) | 113 (99.1) | 0.777 |
| | Moderate acute malnutrition | 22 (0.7) | 21 (0.7) | 1 (0.9) | |
| | Severe acute malnutrition | 12 (0.4) | 12 (0.4) | 0 (0.0) | |
| Baseline CD4 (cells/μl) | ≥500 | 2587 (80.9) | 2495 (80.9) | 92 (80.7) | 1.000 |
| | <500 | 612 (19.1) | 590 (19.1) | 22 (19.3) | |
| Baseline TB status | No signs or symptoms of TB | 3102 (97.0) | 2992 (97.0) | 110 (96.5) | 0.448 |
| | Currently on TB treatment | 51 (1.6) | 49 (1.6) | 2 (1.8) | |
| | Diagnosed with TB | 4 (0.1) | 4 (0.1) | 0 (0.0) | |
| | TB treatment completed | 23 (0.7) | 23 (0.7) | 0 (0.0) | |
| | Presumed with TB | 19 (0.6) | 17 (0.6) | 2 (1.8) | |

participants (2,587 or 80.9%) had CD4 counts ≥500 cells/ul at ART initiation while 3102 (97.0%) had no signs or symptoms of TB. Participants in the intervention and control groups differed with respect to age category (p = 0.001), baseline WHO clinical stage (p = 0.016), baseline weight (p<0.001), and baseline ART regimen (p<0.001).

**Covariate balance between the intervention and control groups before and after applying the optimal bandwidth.** The median VL for the control group was 1 copy/ml (IQR 1–49) and the VL ranged from 0–999 copies/ml. In the intervention group, the median VL was 13,321 copies/ml (IQR 3,890–44,200) and the range was 1030–2,300,000 copies/ml. We estimated an optimal bandwidth of 1824.301 (approximately 912.15 below and 912.15 above) at the VL ≥1,000 copies/ml cut-off using the Imbens-Kalyanaraman approach.

Within this optimal bandwidth, there were 222 participants in the control group (representing 6.9% of people with VL<1000 copies/ml), and 14 participants in the intervention group (representing 12.3% of people with VL ≥1,000 copies/ml). The control and intervention group participants within the optimal bandwidth were similar on observed covariates, with p>0.05 (Table 2).

## Primary and secondary outcome findings before and after optimal bandwidth

When assessing the entire study population (Table 3), repeat VL was suppressed in 3,016 of 3,085 (97.8%) participants in the control group and 93 of 114 (81.6%, p<0.001) participants in the intervention group. Deaths were similar between the control and intervention groups (1.6% versus 0.9%, respectively, p = 0.85). After applying the optimal bandwidth, repeat VL was suppressed in 219 of 222 (97.9%) participants in the control group and 12 of 114 (85.7%, p<0.001) participants in the intervention group. Just one death was observed overall in the control group (p = 1.0).

## Effect of IAC on viral load suppression and all-cause mortality

Analysis of the entire dataset (n = 3,199) showed a lower likelihood of VL suppression in the intervention group compared to the control group in both the unadjusted (OR 0.10; 95% CI 0.06–0.17) and adjusted (aOR 0.12; 95% CI, 0.07–0.21) analyses. There was no difference in mortality when comparing the intervention vs. control groups in both the unadjusted (OR

**Table 2. Covariate balance between the intervention and control groups before and after applying the optimal bandwidth.**

| Covariates | P-values before the optimal bandwidth (n = 3,199) | P-values after the optimal bandwidth (n = 236) |
|---|---|---|
| | P-values | P-values |
| Study site | 0.610 | 0.503 |
| Level of health facility | 0.573 | 0.403 |
| Age (mean (SD)) | <0.001 | 0.532 |
| Sex | 0.601 | 0.899 |
| Point of entry | 0.247 | 0.586 |
| Baseline WHO clinical stage | 0.016 | 0.275 |
| Baseline weight (mean (SD)) | <0.001 | 0.181 |
| Baseline MUAC | 0.777 | 0.192 |
| CD4 category | 1.000 | 0.236 |
| Baseline TB status | 0.448 | 0.458 |

**Table 3. Distribution of study outcomes between the intervention and control groups within optimal bandwidth.**

| Outcome distribution before applying the optimal bandwidth | | | | | |
|---|---|---|---|---|---|
| Variables | Levels | Overall (n = 3,199) | Control group (n = 3,085) | Intervention group (n = 114) | P-value |
| VL suppression | No | 90 (2.8) | 69 (2.2) | 21 (18.4) | <0.001 |
| | Yes | 3109 (97.2) | 3016 (97.8) | 93 (81.6) | |
| Mortality | No | 3150 (98.5) | 3037 (98.4) | 113 (99.1) | 0.848 |
| | Yes | 49 (1.5) | 48 (1.6) | 1 (0.9) | |
| Outcome distribution after applying the optimal bandwidth | | | | | |
| Variables | Levels | Overall (n = 236) | Control group (n = 222) | Intervention group (n = 14) | P-value |
| VL suppression | No | 5 (2.1) | 3 (1.4) | 2 (14.3) | 0.021 |
| | Yes | 231 (97.9) | 219 (98.6) | 12 (85.7) | |
| Mortality | No | 235 (99.6) | 221 (99.5) | 14 (100.0) | 1.000 |
| | Yes | 1 (0.4) | 1 (0.5) | 0 (0.0) | |

0.56; 95% CI 0.08–4.09) and adjusted (aOR 0.39; 95% CI 0.05–2.96) analyses. Similarly, within the optimal bandwidth, the proportion with repeat VL suppression was lower in the IAC group than in the control group (85.7% vs. 98.6%, OR = 0.09, 95% CI, 0.01–0.66) and all-cause mortality did not vary (Table 4).

## Sensitivity analysis results at varying bandwidths

The effect of IAC on VL suppression remained the same (OR = 0.09; 95% CI, 0.01–0.66) when the optimal bandwidth was widened by 25, 50, 75, and 85 both below and above the VL ≥1,000 copies/ml cut-off, and when it was narrowed by 50.

## Discussion

Focusing on PLHIV with an initial VL around the 1000 copies/mL cut-off, repeat VL suppression was lower in those who were above the cut-off and received IAC than in those who were below the cut-off and received routine PSS. These findings suggest IAC is not effective in achieving VL suppression. Our evidence contributes to informing the implementation of IAC to support PLHIV in Uganda and similar settings in sub-Saharan Africa in adhering to ART. Our study is the first to rigorously evaluate the effect of IAC on VL suppression and all-cause mortality in Uganda and sub-Saharan Africa. Consequently, evidence from a similar study design to compare our results is lacking. Nonetheless, previous observational studies conducted in Uganda [5, 8] and Ethiopia [21, 22] all report unsuppressed VL after IAC completion, consistent with our results.

**Table 4. Effect of IAC on VL suppression and all-cause mortality with and without optimal bandwidth.**

| Outcomes | Levels | Unadjusted analysis | Adjusted analysis | RDD analysis |
|---|---|---|---|---|
| | | OR (95% CI) | OR (95% CI) | OR (95% CI) |
| VL suppression | No | 1 | 1 | 1 |
| | Yes | 0.10*** (0.06–0.17) | 0.12*** (0.07–0.21) | 0.09* (0.01–0.66) |
| All-cause mortality | No | 1 | 1 | 1 |
| | Yes | 0.56 (0.08–4.09) | 0.39 (0.05–2.96) | 1 |
| Sample sizes | | 3,199 | 3,199 | 236 |

**Note**: Statistical significance at 5% level: *p<0.05, **p<0.01, ***p<0.001.

Our finding of a lower likelihood of repeat VL suppression despite IAC might be explained by several social and biologically plausible reasons and we attempt to present a few of them. First, IAC is appropriate when unsuppressed VL results from poor ART adherence. Our finding appears to suggest indiscriminate recommendation and application of IAC for all PLHIV with unsuppressed VL regardless of ART adherence status. IAC is ineffective if VL suppression is due to drug resistance but not suboptimal ART adherence.

For instance, among PLHIV on a second-line ART regimen with a high prevalence of resistance in eastern Uganda, IAC failed to reverse unsuppressed VL [5]. Second, a high pre-IAC VL influences VL suppression. Previous epidemiological studies [21, 22] report a higher likelihood of treatment failure (unsuppressed VL after IAC) among PLHIV with a higher pre-IAC VL $\geq 1,000$ copies/ml compared to those with a lower pre-IAC VL $<1,000$ copies/ml. Third, insufficiencies in IAC implementation are prevalent within health facilities. For example, the failure to form a multidisciplinary team to provide IAC has been reported in a previous study in Uganda [7], leading to inadequate ART adherence support or counseling to PLHIV with unsuppressed VL. Another important consideration is the possibility of suboptimal ART adherence during IAC for reasons such as stigma and undisclosed HIV serostatus, for example. In Ethiopia [22], ART adherence remained poor among PLHIV on IAC and subsequently led to a treatment failure. Similar to our setting, evidence indicates that most HIV control programs lack systems to monitor ART adherence during IAC except VL monitoring after IAC completion [23]. Therefore, continuous monitoring of ART adherence during IAC might be needed to detect progress in halting VL increase. Four, IAC as a single intervention might be inadequate to achieve VL suppression. Additional interventions might be needed to complement the role of IAC in addressing the barriers to ART adherence. One study reports that a lack of support from family members or relatives compromises ART adherence leading to unsuppressed VL despite IAC [7]. Lastly, a lack of food or nutritional support to PLHIV on IAC has been reported to compromise ART adherence [7]. The success of IAC might therefore entail nutritional or other structural support to enhance ART adherence.

## Implications of findings

Our findings have ramifications for current practice, policy, and future research. Overall, our findings appear to suggest a need to review and reconsider IAC implementation in supporting ART adherence and achieving VL suppression in Uganda and similar settings. The fidelity of implementation of IAC across HIV control programs to determine the extent to which it is delivered as intended as well as the dose and intensity need rigorous evaluation in future research. Studies further need to examine adherence to the requirements of IAC among PLHIV, including rigorously assessing the reasons for unsuppressed VL despite IAC completion. On the policy front, IAC is largely a health education intervention and is expected to achieve behavioral change for better health outcomes.

However, behavior change is a gradual process so health education without a healthy public policy to tackle structural and environmental barriers to ART adherence such as stigma, drug and substance use, and undisclosed HIV serostatus among others would not lead to the realization of the intended outcomes rather than blaming the person for failure to achieve VL suppression [24].

## Study strengths and limitations

Our study is the first RDD to determine the causal effect of IAC on VL suppression and all-cause mortality in Uganda and sub-Saharan Africa. Our findings remained robust to changing bandwidths at the cut-off and our study has a strong external validity since we analyzed real-world data. The analysis with and without an optimal bandwidth showed a similar effect of

IAC on VL suppression confirming the little benefit to IAC after addressing concerns about selection bias. Limitations of the study included unmeasured confounders due to the use of observational data. We mitigated this limitation by restricting the analysis to PLHIV just below and just above the cutoff where both measured and unmeasured confounders are comparable. Our sample size within the optimal bandwidth is relatively small so studies with a large sample size would be needed to provide additional evidence. By default, our estimate represents the local average treatment effect due to the restrictive nature of the analysis. Lastly, the lack of qualitative data to contextualize the findings is another limitation to consider.

## Conclusions and recommendations

Our study found IAC is not effective in achieving VL suppression among PLHIV on first-line ART in Kampala, Uganda. Our findings suggest a need to investigate the fidelity of IAC implementation and the reasons for VL persistence beyond the suppression threshold.

## Supporting information

**S1 Data. Dataset.**
(DTA)

**S1 File. Inclusivity in global research.**
(DOCX)

## Acknowledgments

We are greatly indebted to the IDI-REC for providing a sound ethical review and approval of the study protocol. We thank the Directorate of Public Health and Management of the Kampala Capital City Authority for their kind administrative support. The health facility leaders of the respective study sites, the HMIS focal persons, and Philip Kalyesubula are much appreciated.

## Author Contributions

**Conceptualization:** Jonathan Izudi, Barbara Castelnuovo, Rachel King, Adithya Cattamanchi.

**Data curation:** Jonathan Izudi, Barbara Castelnuovo, Rachel King, Adithya Cattamanchi.

**Formal analysis:** Jonathan Izudi, Rachel King, Adithya Cattamanchi.

**Funding acquisition:** Jonathan Izudi.

**Investigation:** Jonathan Izudi, Barbara Castelnuovo, Rachel King, Adithya Cattamanchi.

**Methodology:** Jonathan Izudi, Barbara Castelnuovo, Rachel King, Adithya Cattamanchi.

**Project administration:** Jonathan Izudi, Barbara Castelnuovo, Adithya Cattamanchi.

**Resources:** Jonathan Izudi.

**Software:** Jonathan Izudi, Rachel King, Adithya Cattamanchi.

**Supervision:** Barbara Castelnuovo, Rachel King, Adithya Cattamanchi.

**Validation:** Jonathan Izudi, Barbara Castelnuovo, Rachel King, Adithya Cattamanchi.

**Visualization:** Jonathan Izudi, Barbara Castelnuovo, Rachel King, Adithya Cattamanchi.

**Writing – original draft:** Jonathan Izudi, Barbara Castelnuovo, Rachel King, Adithya Cattamanchi.

**Writing – review & editing:** Jonathan Izudi, Barbara Castelnuovo, Rachel King, Adithya Cattamanchi.

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
