## [Decision Letter · Decision Letter 0]

22 May 2023

PGPH-D-23-00651

Impact of intensive adherence counseling on viral load suppression and mortality among people living with HIV in Kampala, Uganda: a regression discontinuity design

Dear Dr. Izudi,

Thank you for submitting your manuscript to PLOS Global Public Health. After careful consideration, we feel that it has merit but does not fully meet PLOS Global Public Health’s publication criteria as it currently stands. Therefore, we invite you to submit a revised version of the manuscript that addresses the points raised during the review process.

We look forward to receiving your revised manuscript.

Kind regards,

Henry Zakumumpa, PhD

Academic Editor

Journal Requirements:

Additional Editor Comments (if provided):

Reviewers' comments:

Reviewer's Responses to Questions

**Comments to the Author**

1. Does this manuscript meet PLOS Global Public Health’s publication criteria? Is the manuscript technically sound, and do the data support the conclusions? The manuscript must describe methodologically and ethically rigorous research with conclusions that are appropriately drawn based on the data presented.

Reviewer #1: Yes

Reviewer #2: Yes

2. Has the statistical analysis been performed appropriately and rigorously?

Reviewer #1: Yes

Reviewer #2: Yes

3. Have the authors made all data underlying the findings in their manuscript fully available (please refer to the Data Availability Statement at the start of the manuscript PDF file)?

Reviewer #1: Yes

Reviewer #2: No

4. Is the manuscript presented in an intelligible fashion and written in standard English?

Reviewer #1: Yes

Reviewer #2: Yes

5. Review Comments to the Author

Reviewer #1: Overall, this is a well-written paper, therefore, my review outcomes are suggestions some of which may not be adopted should the authors see it fit.

Abstract: Overall, the abstract provides a clear and concise summary of the study. Here are some specific comments for improvement:

• The abstract does not provide enough context for the problem being studied. It would be helpful to provide some background information on the issue of HIV and ART adherence, as well as the prevalence of VL ≥1,000 copies/ml in the study population.

• The abstract could benefit from a clearer description of the study design and methods. Specifically, it would be helpful to explain the regression discontinuity design in layman's terms and why it was appropriate for this study.

• The results are clearly presented, but it would be helpful to provide some explanation for why the odds of repeat VL suppression were lower in the intervention group. Were there any potential reasons for this finding that were discussed in the study? Additionally, it would be helpful to mention the effect size of the intervention (i.e., the actual difference in VL suppression rates between the control and intervention groups).

• The abstract ends with some useful recommendations for future research, but it would be helpful to provide some more context for why investigating the fidelity of IAC implementation and reasons for VL persistence are important.

Background: The background provides a clear overview of the issue of viral load suppression in people living with HIV (PLHIV) and the use of intensive adherence counseling (IAC) to improve adherence to ART. It summarizes existing evidence on the effectiveness of IAC, highlighting mixed results from previous studies. The authors note the lack of causal evidence on the effect of IAC on viral load suppression, which the EFFINAC study aims to address. Overall, the background provides relevant and concise information to contextualize the study and the research question. However, some additional information on the importance of viral load suppression and the impact of unsuppressed viral loads on PLHIV's health outcomes could further strengthen the background.

Study design: The study design presented, regression discontinuity design (RDD), appears appropriate for evaluating the causal effect of IAC on VL suppression and all-cause mortality. The use of RDD is justified since intervention assignment depends on a cut-off, with IAC being provided to PLHIV with unsuppressed VL (VL ≥1,000 copies/ml), while those with suppressed VL receive routine psychosocial support (PSS). RDD is suitable when there is a discontinuity in the probability of treatment assignment at a certain threshold, and participants just above and below this threshold are comparable. The study appears to meet the RDD conditions, with VL serving as the assignment variable, the cut-off being VL ≥1,000 copies/ml, and outcomes being measured for all participants regardless of intervention.

Study population and other sections through measurements: Overall, the description of the study population and measurements is comprehensive and clear

Study limitation: Could this be a potential limitation of the study design; that it assumes that individuals just above and below the cut-off are similar in both measured and unmeasured confounders, which may not always be the case. Additionally, the study design cannot account for potential selection bias, where individuals with different characteristics are more likely to receive IAC based on other factors not accounted for in the RDD. Therefore, it is important to interpret the findings of the study with caution and consider these potential limitations in the interpretation of the results.

Discussion: Overall, this discussion provides a well-organized and informative critique of the effectiveness of IAC in achieving VL suppression among PLHIV in Uganda and similar settings in sub-Saharan Africa. The discussion cites relevant studies and provides plausible explanations for the lower likelihood of repeat VL suppression despite IAC. Additionally, the implications of the findings for practice, policy, and future research are well-articulated. However, there are a few areas where the discussion could be improved. Firstly, the discussion could benefit from a more detailed explanation of what IAC is and how it works to support ART adherence. This would help readers who are not familiar with the intervention to better understand the findings and implications of the study. Secondly, the discussion could have been more specific in identifying the limitations of the study, such as the small sample size, the lack of randomization, and the potential for confounding variables. By acknowledging these limitations, the authors could have enhanced the credibility and transparency of their study. Finally, the discussion could have provided more concrete recommendations for how IAC implementation could be improved. While the authors suggest that the fidelity, dose, and intensity of IAC implementation need rigorous evaluation in future research, they do not provide specific suggestions for how these improvements could be made. Providing more specific recommendations could help guide policy and practice in a more tangible way. Overall, while the discussion provides a thorough and insightful critique of the study's findings, there are opportunities for improvement in terms of providing more detailed explanations, acknowledging limitations, and offering specific recommendations for future research and practice.

Reviewer #2: I did not find any statement about availability of data. Otherwise the manuscript highlights an important concern in HIV treatment and care. It is well written and all analyses are well done. What is not clear is how the participants in the control group that had unsuppressed viral load at the end of the study were handled. The inclusion criteria is also silent about the duration on ART for the participants included in the study. My major concern is how receiving IAC was measured to conclude that all participants in the intervention arm received at least 3 sessions of IAC, yet this being a retrospective data analysis using routinely collected data. I would expect some level of missingness both in IAC and viral load results.

Thank you

6. PLOS authors have the option to publish the peer review history of their article (what does this mean?). If published, this will include your full peer review and any attached files.

**Do you want your identity to be public for this peer review?** For information about this choice, including consent withdrawal, please see our Privacy Policy.

Reviewer #1: No

Reviewer #2: **Yes: **Bernard Kikaire

---

## [Editor Report · Decision Letter 1]

6 Jun 2023

PGPH-D-23-00651R1

Impact of intensive adherence counseling on viral load suppression and mortality among people living with HIV in Kampala, Uganda: a regression discontinuity design

Dear Dr. Izudi,

Thank you for submitting your manuscript to PLOS Global Public Health. After careful consideration, we feel that it has merit but does not fully meet PLOS Global Public Health’s publication criteria as it currently stands. Therefore, we invite you to submit a revised version of the manuscript that addresses the points raised during the review process.

We look forward to receiving your revised manuscript.

Kind regards,

Henry Zakumumpa, PhD

Academic Editor

Journal Requirements:

Additional Editor Comments (if provided):

We are delighted to share with you reports from our reviewers. Please pay attention to suggestions for improving your discussion section.
---

## [Decision Letter · Decision Letter 2]

14 Jul 2023

Impact of intensive adherence counseling on viral load suppression and mortality among people living with HIV in Kampala, Uganda: a regression discontinuity design

PGPH-D-23-00651R2

Dear Jonathan Izudi,

We are pleased to inform you that your manuscript 'Impact of intensive adherence counseling on viral load suppression and mortality among people living with HIV in Kampala, Uganda: a regression discontinuity design' has been provisionally accepted for publication in PLOS Global Public Health.

Best regards,

Henry Zakumumpa, PhD

Academic Editor

Thank you for addressing the comments from our reviewers.

Reviewer Comments (if any, and for reference):

Reviewer's Responses to Questions

**Comments to the Author**

1. If the authors have adequately addressed your comments raised in a previous round of review and you feel that this manuscript is now acceptable for publication, you may indicate that here to bypass the “Comments to the Author” section, enter your conflict of interest statement in the “Confidential to Editor” section, and submit your "Accept" recommendation.

Reviewer #1: All comments have been addressed

Reviewer #2: All comments have been addressed

2. Does this manuscript meet PLOS Global Public Health’s publication criteria? Is the manuscript technically sound, and do the data support the conclusions? The manuscript must describe methodologically and ethically rigorous research with conclusions that are appropriately drawn based on the data presented.

Reviewer #1: Yes

Reviewer #2: Yes

3. Has the statistical analysis been performed appropriately and rigorously?

Reviewer #1: Yes

Reviewer #2: Yes

4. Have the authors made all data underlying the findings in their manuscript fully available (please refer to the Data Availability Statement at the start of the manuscript PDF file)?

Reviewer #1: Yes

Reviewer #2: Yes

5. Is the manuscript presented in an intelligible fashion and written in standard English?

Reviewer #1: Yes

Reviewer #2: Yes

6. Review Comments to the Author

Reviewer #1: Thank you Izudi and Colleagues for this great piece of writing. I hope it will be a critical resource for IAC programming.

Reviewer #2: The manuscript is now clear and authors have addressed all the review comments adequately.

7. PLOS authors have the option to publish the peer review history of their article (what does this mean?). If published, this will include your full peer review and any attached files.

**Do you want your identity to be public for this peer review?** For information about this choice, including consent withdrawal, please see our Privacy Policy.

Reviewer #1: **Yes: **Kimuli Derrick

Reviewer #2: **Yes: **Bernard Kikaire
